# Gender differences in adverse childhood experiences, resilience and internet addiction among Tunisian students: Exploring the mediation effect

Imen Mlouki[1,2,3], Mariem Majdoub[2], Emna Hariz[2]*, Ahlem Silini[1,2], Houcem Elomma Mrabet[4], Nejla Rezg[2], Sana El Mhamdi[1,2,3]

1 Preventive and Community Medicine, Faculty of Medicine of Monastir, University of Monastir, Monastir, Tunisia, 2 Department of Preventive and Community Medicine, Taher Sfar University Hospital, Mahdia, Tunisia, 3 Research Laboratory "Epidemiology Applied to Maternal and Child Health" 12SP17, Monastir, Tunisia, 4 Department of Medicine, Taher Sfar University Hospital, Mahdia, Tunisia

* emna.harizz@gmail.com

**Data Availability Statement:** The data used to support the findings of the study are available in the supporting information file.

## Abstract

Adverse Childhood Experiences (ACEs) are a common public health issue with a variety of consequences, including behavioral addiction such as Internet Addiction (IA). Despite widespread recognition of this issue, the underlying mechanisms are not well studied in recent literature. Additionally, studies have indicated gender disparities in the prevalence and manifestation of ACEs and IA. The objective of this study was to investigate the mediating effect of resilience on the link between ACEs and IA among high-school students according to gender in Mahdia city (Tunisia). We conducted a cross-sectional survey for two months (January- February 2020), among 2520 schooled youth in Mahdia city (Tunisia). The Arabic-language edition of the World Health Organisation ACE questionnaire was used. The validated Arabic versions of the Adolescent Psychological Resilience Scale and the Internet Addiction Test were the screening tools for resilience and IA. Data were analyzed according to gender. The majority of youth (97.5%) were exposed to at least one ACE with the most prevalent being emotional neglect (83.2%). Exposure to extra-familial ACEs was also high reaching 86.9% with higher rates among boys for all types of social violence. Internet addiction was common among students (50%) with higher prevalence for boys (54.4% vs 47.7% for girls, p = 0.006). Resilience scores were 86.43 ± 9.7 for girls vs 85.54 ± 9.79 for boys. The current study showed that resilience mediated the link between ACEs, especially intrafamilial violence, and internet addiction (%mediated = 15.1). According to gender, resilience had a significant mediating role on internet addiction for girls (%mediated = 17) and no significant role for boys. The mediating effect of resilience in the relationship between ACEs and cyber-addiction among schooled adolescents in the region of Mahdia (Tunisia) has been identified.

**Funding:** The authors received no specific funding for this work.

**Competing interests:** The authors have declared that no competing interests exist.

## Introduction

Adverse Childhood Experiences (ACEs), which include being subject to physical, mental or sexual violence as well as extra-familial adversities during the first 18 years of life, are connected to a wide range of detrimental health impacts in adolescence and later life [1, 2]. According to recent literature, there are gender disparities concerning childhood maltreatment [3]. ACEs have been linked to a variety types of addiction, not only substance use disorders but also behavioral addictions such as excessive use of internet [4]. According to the World Health Organization (WHO) [5], cyberaddiction is becoming an alarming phenomenon with globally high prevalence rates, predicting an increasing impact with long-lasting psychological and social dangers. In Tunisia, a 2019 survey showed that 43.9% of young students were addicted to internet [6]. Moreover, a Chinese study in 2022, reported that cyberaddiction and mechanisms underlying its associated factors were different according to gender [7].

The mechanism through which ACEs impact teenage behavior have been extensively investigated [2, 8]. However, few studies examined the pathway between ACEs and internet addiction in recent literature [9]. Several regulating virtues have been reported as protective factors against smartphone addiction such as temperance as well as courage [10]. Resilience, broadly defined as the ability of a person to adapt to adverse situations in a positive manner [11, 12], has been recently identified as one of the key protective factors. Indeed, a 2018 study conducted among middle-school youth in Korea revealed the buffering effect of resilience toward internet addiction in girls [13]. Furthermore, research has shown that there is a significant relationship between adverse childhood experiences (ACEs) and resilience, with studies indicating that early life adversities had a negative association with individual resilience [14, 15]. In addition, a recent research carried out in Iran showed that university students with high resilience scores were less prone to substance use despite their vulnerability to addiction [11]. In Slovakia, 2021, it was found that resilience decreased the probability of Emotional and Behavioural Problems (EBP) among adolescents, with documented resilience mediation noted in the relationship between ACE and EBP among teenagers [16]. It was also revealed that resilience played a protective role among psychologically maltreated youngsters in Turkey [17]. Recently, it was found that the gender plays a role in shaping resilience [18]. In Tunisia, a recent ACE survey among youth revealed alarming rates. Indeed, 99.1% reported intra-familial ACEs and 84% expressed being victim of social adversities [2]. Despite these high rates, there is a gap in the Tunisian literature regarding ACEs, let alone ACEs and resilience.

Giving these data and the scarcity of literature in this context, it seems crucial to examine the role of resilience as a contributing element linking ACEs and internet addiction among students in Tunisia in order to establish more efficient and targeted prevention plans among this critical population. Thus, we aimed at assessing the relationship between ACEs and cyberaddiction mediated by resilience among schooled youngsters in Mahdia City, Tunisia.

## Methodology

### Population study and selection

Participants schooled in all secondary colleges of Mahdia governorate were recruited from January to February 2020. Mahdia is a Tunisian city with a population of 441989 (as at the 2020 census) [19] and a gross enrollment ratio of 94.2% [20].

We randomly selected four classes from each secondary school based on cluster sampling. If we consider a probability of type one error ($\alpha$) equal to 0.05, a precision of 3% and an internet addiction rate of 43.9% [21], the minimal sample size needed for the current survey is 1052 students.

**Inclusion criteria/exclusion.** We included all the consented schooled adolescents from Mahdia and Gafsa cities and we excluded students who consent to complete the questionnaire and who did not return it.

A total of 2520 schooled youth were recruited and 1940 returned the questionnaires with an overall response rate of 77%.

## Measurement tools

**Measurement of adverse childhood experiences.** The **Adverse childhood experiences-International Questionnaire** (ACE-IQ) developed by WHO [8] was used. The Arabic version was validated in Saudi Arabia [22]. The ACE-IQ is intended to measure family dysfunction, physical, sexual and emotional abuse and neglect as well as exposure to bullying, community or collective violence [8]. The ACE score contains nine items (Six items on intrafamilial ACEs and three others on extra familial ACEs). Each item was given a zero or a one point. Thus, the ACE score was marked out of nine.

**Evaluation of internet addiction.** Cyberaddiction was screened via the validated Arabic version of the **Internet Addiction Test** (IAT) [23]. Indeed, the internal consistency and the reliability of this scale was excellent (Cronbach alpha coefficient = 0.921). Total IAT scores ranging from 40 to 69 represents over-users with frequent problems caused by their internet use, and scores between 70 and 100 represents cyberaddiction [24]. Thus, the higher the score is, the higher is the level of internet addiction [25].

**Measurement of resilience.** As defined by the American Psychological Association [26], resilience is the process of adapting well in the face of adversity or significant sources of stress such as relationship problems, serious health problems or workplace and financial stressors. Resilience was measured in the current study using the validated Arabic version of the **Adolescent Psychological Resilience Scale** [27]. In fact, the exploratory factor analyses of its construct validity explained 56.9% of the total variance and the Cronbach alpha coefficient ranged from 0.61 to 0.89 for its different subscales.

The score assesses six sub-dimensions covering three intrapersonal factors (Adjustment, Sense of struggle, empathy) and three interpersonal factors (Family support, Confidant-friend support, School support). The higher the score is, the higher is the level of resilience [27].

**Statistical analysis.** Data analyses were performed via SPSS; Version 21. Categorical variables were represented using frequencies and numeric variables were represented using means and standard deviations (SD). To compare percentages and means according to gender, we used the Chi square and the Student tests, respectively. A p-value less than 0.05 was considered statistically significant. ACE, Resilience, Impulsivity and Internet Addiction were considered as quantitative variables. We verified the distribution of these variables, if they followed the normal distribution, the Pearson correlation test was used. If not, Spearman correlation test was used.

**Mediation analysis.** In this analysis, we examined resilience as a continuous variable to assess its potential role as a mediator. Correlation tests were utilized to determine the zero-order associations between ACEs, resilience and internet addiction. The correlation test was used to check the sign of the correlation coefficient r. If r< 0, the direction of the relationship between the variables is negative. If r> 0, the direction of the relationship between the variables is positive.

Mediation modeling was conducted to ascertain the existence of a significant mediation (or indirect effect) of resilience in the association between ACEs and internet addiction. Resilience was regarded as a potential mediating variable when incorporating it into the model led to a partial or complete reduction of the relationship between ACE as the explanatory variable and internet addiction as the dependent variable [28]. Mediation analyses were carried out utilizing the PROCESS macro created by Andrew F. Hayes [29].

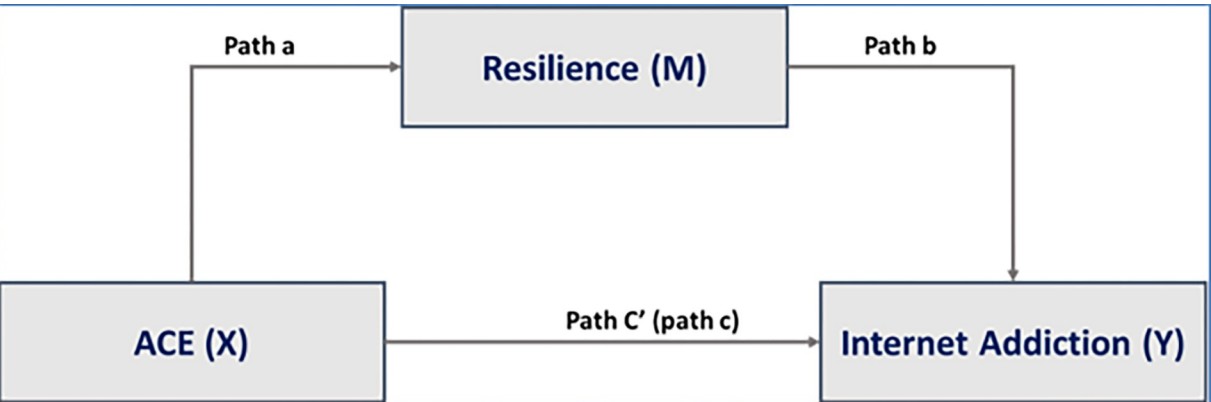

**Fig 1. Diagram of the theoretical link between ACE and internet addiction through resilience.**

Three major conditions need to be met for mediation analyses. Firstly, the link between the explanatory variable (ACE) and the dependent variable (internet addiction) has to be verified (pathway c). Secondly, the mediation variable (resilience) has to be significantly linked to internet addiction (pathway b). Finally, the link between ACE and resilience must be significant (pathway a). Mediation effect occurs when pathway c is significantly reduced [partial mediation) or no longer significant (full mediation) when the mediator is included into the pathway c (pathway c ′) (Fig 1).

**Ethical concerns.** The Ethic Committee of Mahdia Teaching Hospital accepted the protocol of the current survey with an approval number of P01 M.P.C- 2020.

The questionnaire was anonymous and self-administered by students. Physicians were in classrooms to elucidate the intent of the survey to students.

The participation was voluntary. In fact, having the writing consent was impossible since the data collection was randomized but informed consent was taken from parents and students before collecting data. In fact, we contacted headmasters before the survey, in order to inform parents and students in schools.

## Results

### General characteristics of schooled youngsters in Mahdia city

A total of 1940 returned the questionnaires with a mean age of 17 ±1.5. Girls represented of the majority of the study sample (66.3%). At the first level of education, men were more prevalent than women (32% vs. 27.6%, p 0.01). The fourth level, however, had a higher percentage of girls (25.7% as opposed to 20.5%).

### Distribution of ACEs by gender within and outside of families

Table 1 presents the breakdown of all categories of ACEs. In fact, intra-familial ACEs (94.5%) were more reported by students than social ACEs (86.9%). Boys were significantly more exposed to physical abuse and all types of social ACEs than girls (Table 1).

### Resilience score results and internet addiction prevalence among students by gender

Table 2 summarizes the total score of the adolescent psychological resilience scale as well as scores on its six sub-dimensions by gender. Resilience scores were moderate to high among

**Table 1. Prevalence of ACEs among students by gender.**

| Categories of ACEs, n (%) | Total (n = 1940) | Boy (n = 662) | Girl(n = 1278) | p value |
|---|---|---|---|---|
| **Intra-familial ACEs** | 1786 (94.5) | 608 (95.8) | 1178 (93.9) | 0.2 |
| Emotional neglect | 1601 (83.2) | 538 (83.8) | 1057 (83.1) | 0.6 |
| Household dysfunction | 1534 (80.5) | 499 (78.5) | 1030 (81.7) | 0.09 |
| Physical abuse | 1082 (56.3) | 391 (**61**) | 687 (54.1) | **0.004** |
| Emotional abuse | 594 (30.8) | 197 (30.6) | 394 (31) | 0.8 |
| Physical neglect | 469 (24.4) | 173 (26.9) | 293 (23.1) | 0.06 |
| Sexual abuse | 257 (13.4) | 93 (14.5) | 162 (12.8) | 0.2 |
| **Social ACEs** | 1648 (86.9) | 589 (**92.5**) | 1059 (84) | **<0.0001** |
| Community violence | 1417 (73.5) | 541 (**84**) | 870 (68.3) | **<0.0001** |
| Peer violence | 1224 (64.1) | 437 (**68.5**) | 784 (62.1) | **0.006** |
| Collective violence | 359 (18.6) | 188 (**29.2**) | 169 (13.3) | **<0.0001** |

questioned students with a higher total resilience score among girls. Regarding sub-dimension scores, girls had a significantly higher score for school support (15.53 ±3.63 vs 14.84 ±3.7, p<0.001) (Table 2).

## Resilience as a mediator between ACEs and internet addiction among students

Every variable incorporated in the mediation analysis displayed significant correlations. (Table 3). The total count of ACEs **(pathway c)** and Resilience **(pathway b)** were independently linked with Internet addiction (p<0.001), and ACEs were significantly linked with Resilience **(pathway a)** (p <0.001);the number of total ACEs and resilience were inversely correlated (p< 0.001, r = -0.34) (when the number of total ACEs increased, resilience decreased), oppositely, total ACEs and IA score were positively correlated (p< 0.001, r = 0.30) (when the number of ACEs increased, IA increased). Besides, we found that Resilience and IA scores were negatively correlated (p< 0.001, r = -0.19) (when resilience decreased, IA increased).

The total count of all ACEs, which served as the exposure variable, and internet addiction, which served as the outcome variable, both showed statistically significant partial mediation effects of resilience (p< 0.001, % mediated = 10.3%) (Pathway c'). In fact, resilience contributes by 10.3% as a protective factor in the link between ACEs and IA.

Resilience was found to partially mediate the impact of both intra-familial ACEs and extra-familial ACEs. Intra-familial ACEs exhibited a higher degree of mediation by Resilience,

**Table 2. Resilience scores among schooled youngsters by gender (n = 1887).**

| Characteristics | Boy (n = 634) | Girl (n = 1253) | p value |
|---|---|---|---|
| Total resilience score (min = 50, max = 109) | 85.54 ± 9.79 | **86.43 ± 9.7** | **0.063** |
| **Sub-dimension scores** | | | |
| Family support | 22.43 ± 4.18 | 22.72 ± 4.43 | 0.16 |
| Confidant / Friend support | 15.73 ± 3.84 | 15.64 ± 4.12 | 0.66 |
| School support | 14.84 ± 3.7 | **15.53 ± 3.63** | **<0.0001** |
| Adjustment | 10.3 ± 2.35 | 10.16 ± 2.31 | 0.22 |
| Sense of struggle | 13.06 ± 2.08 | 13.05 ± 2.16 | 0.91 |
| Empathy | 9.24 ± 2.23 | 9.33 ± 2.15 | 0.36 |

Fifty percent of youth had cyberaddiction in our sample. Boys were more likely to be addicted than girls (54.4% vs 47.7%, p< 0.006).

**Table 3. Zero-order relationships between ACE, resilience and internet addiction among students.**

|  | (1) | (2) |
|---|---|---|
| **Number of total ACEs** | -0.34*** | 0.30*** |
| (1) Resilience | – | -0.19*** |
| (2) Internet addiction | – | – |
| **Number of intrafamilial ACEs** | -0.32*** | 0.25*** |
| **Number of extrafamilial ACEs** | -0.25*** | 0.27*** |

***: p < 0.001

accounting for 15.1% of the effect (p < 0.001), whereas exposure to extra-familial ACEs showed a slightly lower mediation effect at 10.1% (p <0.001) (Pathway c') (Table 4).

According to gender (Table 5), Among girls, there were noteworthy interactions observed between childhood adversities and resilience in relation to internet addiction (p < 0.001; % mediation = 17%). However, for boys, no statistically significant indirect effect was detected for ACEs in their association with internet addiction through resilience (p = 0.83; % mediation = 0.08%) (Table 5).

## Discussion

Our survey clearly highlights the mediating role of resilience between ACEs, especially intrafamilial violence and internet addiction among students in Mahdia city (% mediation = 15.1). According to gender, we found higher mediation part among girls (p <0.001; % mediation = 17). However, no significant indirect effect was found between ACEs and cyberaddiction through resilience among boys (p = 0.83; % mediation = 0.08%). To the best of our understanding, this is the initial survey in the Middle-East and North Africa region that explores this pathway.

The studied sample counted 1940 schooled adolescents of which girls represented 66.3%. We observed that while boys were more widespread than girls at the initial educational stage, they were less common in the fourth tier. This difference may be clarified by social and economic factors leading to higher drop-out rates for male adolescents. Our results are not consistent with those reported in a Malysian review [30]. This might be related to our retrospective data collection. In view of the scarcity of Tunisian literature about drop-out among youth, a prospective national study will be useful to explore this phenomenon.

We found that 94.5% and 87% of schooled adolescents reported experiencing intrafamilial and social adversities respectively. These frequencies are high compared to other developing non-westernized countries, for example a research in china showed50% of exposure to ACEs [31]. Similar rate(52.6%) was reported by youth in eight European countries including

**Table 4. Mediation model of the relationship of ACE's types on internet addiction with resilience as a mediator among youth (N = 1830).**

| Mediator[‡] | Coefficients* | | | | Sobel test | | % Mediated[†] |
|---|---|---|---|---|---|---|---|
|  | a | b | C | c' | SE | P |  |
| Type of ACEs: | | | | | | | |
| **Total ACEs** | -1.72 | -0.21 | 3.41 | 3.05 | 0.09 | <0.001 | 10.3 |
| **Intra-familial ACEs** | -2.26 | -0.27 | 3.95 | 3.33 | 0.12 | <0.001 | **15.1** |
| **Social ACEs** | -2.69 | -0.28 | 6.79 | 6.03 | 0.15 | <0.001 | 10.1 |

[†] % Mediated = c−c'/c, ‡Mediator: Resilience Adjusted to gender and age

**Table 5. The association between ACEs and internet addiction through resilience among students according to gender.**

| Mediator[‡] | a | b | Coefficients* | | Sobel test | | % Mediated[†] |
|---|---|---|---|---|---|---|---|
| | | | c | c' | SE | p | |
| Mediation model for girls (N = 1216) | | | | | | | |
| Total ACEs | -1.79 | -0.30 | 3.27 | 2.71 | 0.12 | **<0.001** | **17%** |
| Mediation model for boys (N = 607) | | | | | | | |
| Total ACEs | -1.59 | -0.01 | 3.47 | 3.44 | 0.14 | 0.83 | 0.08% |

┼ % Mediated = c–c'/c

‡Mediator: Resilience

Albania, Latvia, Lithuania, Montenegro, Romania, the Russian Federation, North Macedonia, and Turkey [32]. Our results are closer, while still higher, to those found in Saudi Arabia (81.7%) [22] and in Vietnam (76.2%) [33]. A Tunisian study showed that exposure to intra-familial violence was more reported than social adversities with 99.1% and 84% respectively [34]. Thus, these alarming rates call for urgent measures to be taken in order to mitigate leading factors to ACEs.

Based on our findings, the most reported intrafamilial violence among youth was being emotionally neglected (83.2%) succeeded by household dysfunction (80.5%) with no gender difference. The ranking is aligned with western results despite an important difference between rates. For instance, a Japanese study among students revealed that the highest intra-familial ACEs were household dysfunction (13.8%) and emotional abuse (13.7%) with a higher prevalence among girls (16% vs 11.6%) [35]. Similarly, female adolescents in the United Kingdom were more exposed to emotional abuse (8% vs 5.5%) [36]. These differences may be explained by social and cultural determinants. Another strikingly high rate is that 13.4% of students reported being sexually abused with no gender difference. It is important to note that this percentage may be underestimated due to disclosure problems and socio-cultural factors. According to a recent meta-analysis, the global rate of child sexual abuse is around 7% for boys and 19% for girls [37]. In our sample, the rate for boys (14.5%) is double the global rate while the rate for girls is relatively lower. Our findings are concordant with those reported in Croatia (10.8%) [38] and Saudi Arabia(14%) [39]. This substantial contrast between rates might be related to the dissimilarity in measurement instruments as well as the scarcity in data especially in conservative countries. Indeed, conducting studies at a broader scale and taking immediate actions to prevent this alarming phenomenon are needed. According to the Center for Disease Control, ACEs primary prevention is based on strengthening household financial security, establishing "family-friendly work policies" and connecting youth to caring adults and activities" in after-school programs [40].

Regarding social violence, being victim of bullying was common (64.1%) among questioned adolescents. This prevalence is significantly higher compared to rates observed in the United States (41.2%) [41] and in Arabic countries such as Algeria (38.3%) and Lebanon (25.1%) [42]. More particularly, cyberbullying is a growing worldwide concern related to the growing use of social networks applications especially among youth. In our sample, 33.4% of schooled adolescents reported being victims of cyberbullying. The Global Kids online study [43] revealed similar cyberbullying rates in Bulgaria (38%) and in Chile (36%) whereas rates were significantly higher in Uruguay (56%) and in Italy (48%). A survey conducted among young people in 42 countries about Social Media Use (SMU) showed that cyberbullying victimization was higher for girls (17.2% vs 13.9%) [44]. This highlights the negative facet of unsupervised and uneducated internet use among adolescents. In fact, it was demonstrated that cyberbullying lead to

several mental health issue including depression, borderline personality disorder, sleep deprivation and suicidal thoughts [45]. A global approache in forcing protective behaviors among youth but also supervising online content is required by collaborating with technology sponsors, social actors and educational staff [46]. In addition, it is advisable to take in charge victims early with clear instructions and guidance.

One of the key observations in our study is the prevalence of internet addiction (50%) with a higher prevalence among boys. Our finding is consistent with a 2019 Tunisian survey [6] showing that 49.9% of students were addicted. These rates are extremely high compared to those shown by a meta-analysis including 30 studies from Europe, Asia, America and Oceania and estimating an overall addiction prevalence of 24.6% among youth [47]. It is important to note that these risky behaviors suspected to be accentuated during the COVID-19 outbreak. Indeed, cyberaddiction rate among students climbs from 11.69% [48] to 24.4% [49] after the pandemic in Taiwan. Interned addiction is recognized to have both immediate and long-term consequences such as sleep disorders [50], social isolation [51], active violence especially amongst boys and suicidal ideation [38, 52].

To face the widespread of internet addiction and its harmful side effects among youth, it is also useful to investigate mediators that help preventing this risky behavior. According to a recent meta-analysis [53], intrapersonal determinants (self-identity, self-control, emotional regulation..) had higher protective effect than interpersonal variables (relational ability, family relationship..). In our study, we evaluated resilience among schooled adolescents based on both intrapersonal factors covering adjustment, sense of struggle as well as empathy and interpersonal factors including family support, friend support and school support. We found that resilience scores were moderate to high among youth with a score of 86.43 ± 9.7 for girls vs 85.54 ± 9.79 for boys. Our findings are in line with those reported among adolescents in Iran (84.41±11.01) and in Poland (67.66 ±15.15) [54, 55]. Gender disparity was identified in both studies: In Iran, the resilience level was significantly higher among girls. However, girls had significantly lower levels in Poland [54, 55].

Our research reveals the significant mediating effect of resilience in the link between exposure to ACEs and internet addiction among adolescents (% mediated = 10.3). Regarding the association between ACEs and resilience, a2022 study [56] showed similar results. In fact, cumulative ACE exposure was associated with lower resilience among adult women in Iceland [56]. Concerning the relationship between resilience and internet addiction, several western studies investigated this link. However, few data were available in developing countries and we were unable to compare our results. Our finding is in agreement with a recent study examining the protective role of resilience [57] and showing that higher resilience score was associated with lower internet addiction level. Added to that, it was proven in China that enhancing children resilience can be an effective way to reduce cyberaddiction [58]. According to gender, this mediation was only significant for girls (%mediated = 17). Our interesting result is concordant with a Korean survey [13] concluding that, although there was no significant sex difference in resilience, the buffering effect of resilience toward internet addiction only emerged in girls. This fact may be inferred by the difference in goals and motivation for internet use between boys and girls. Indeed, it was reported that boys are more likely to use the internet for pleasure [4] by engaging in online gaming or cybersexual activities [52] whereas girls are more likely to use internet to seek information and engage in social networking [59]. Thus, a potential explanation to the major mediation effect of resilience for girls, is that benefiting from stronger interpersonal resilience factors (Family support, friend support, school support) can balance the primary motivation of girls for internet use (information seeking and social networking) and hence reduce the risk of internet addiction [13]. However, this mediating effect is more limited for boys who have predominantly different motivation.

These results emphasize the importance of implementing effective measures to reinforce resilience by targeting both intrapersonal and interpersonal factors. It is compulsory to promote safe home environment by addressing issues such as neglect and household dysfunction. Providing access to mental support systems and involving extracurricular activities that raise awareness about ACEs as well as the protective role of resilience are needed [60–62]. These recommendations should be adapted to fit the resources of low-income communities.

The current study has some limitations. First, the cross-sectional nature of data collection did not provide an understanding of the temporal association between adverse experiences, resilience, and internet addiction. Second, cyberaddiction was evaluated globally, with no type specification (gaming, social media, information seeking). Consequently, additional future studies using targeted measurement tools are needed to better determine protective and risk factors. Third, the present survey was carried out in a single Tunisian governorate, which limits the representativeness of our findings. In addition, the rate of cyberaddiction among students has recently increased after thecovid-19 pandemic [49]. Thus, reported rates in our research are probably underestimated. Evaluating the impact of covid-19 outbreak on risky behaviors is need. Fourthly, we notice that the resilience scale is too long and contains similar items that maybe confusing for the respondent. However, to the best of our knowledge, it was the only available validated Arabic tool for screening resilience. Finally, in the mediation model, we have not adjusted for confounding variables such as depression and anxiety. Future studies need to consider these variables.

## Conclusion

Our study is among the initial ones in Tunisia to investigate the mediating role of resilience in the association between ACEs and cyberaddiction in the adolescent population. Specifically, we found a major contribution of resilience between intra-familial ACEs and internet addiction for girls (%mediation = 17) with no significant indirect effect for boys. This opens the door to child and adolescent psychiatrists to explore this pathway and urgently implement effective preventive strategy based on developing life skills and promoting safe use of internet among youth in Tunisia.

## Supporting information

**S1 Data. Adverse childhood experiences, resilience and internet addiction.**
(SAV)

**S2 Data. The full list of legends for the supporting information file.**
(DOCX)

## Acknowledgments

The authors are grateful to the following people for their help in data collection: Faouzia Chebbi, Sarra Nouira, Ines Daldoul Chebbi, Fethi Aroui and Asma Sayadi. We seize this opportunity to express our gratitude to all the team of The Epidemiology and Preventive Medicine Department at the University Hospital Taher Sfar Mahdia.

## Author Contributions

**Conceptualization:** Imen Mlouki, Houcem Elomma Mrabet, Sana El Mhamdi.

**Data curation:** Imen Mlouki, Mariem Majdoub, Emna Hariz, Ahlem Silini, Houcem Elomma Mrabet, Nejla Rezg, Sana El Mhamdi.

**Investigation:** Imen Mlouki, Mariem Majdoub, Houcem Elomma Mrabet.

**Methodology:** Imen Mlouki, Sana El Mhamdi.

**Supervision:** Imen Mlouki, Sana El Mhamdi.

**Writing – original draft:** Imen Mlouki, Mariem Majdoub, Emna Hariz.

**Writing – review & editing:** Imen Mlouki, Ahlem Silini.

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
