## [Decision Letter · Decision Letter 0]

12 Sep 2023

PGPH-D-23-01242

Gender differences in adverse childhood experiences, resilience and internet addiction among Tunisian students: Exploring the mediation effect.

Dear Dr. Hariz,

Thank you for submitting your manuscript to PLOS Global Public Health. After careful consideration, we feel that it has merit but does not fully meet PLOS Global Public Health’s publication criteria as it currently stands. Therefore, we invite you to submit a revised version of the manuscript that addresses the points raised during the review process.

We look forward to receiving your revised manuscript.

Kind regards,

Humayun Kabir

Academic Editor

Journal Requirements:

a. State what role the funders took in the study. If the funders had no role in your study, please state: “The funders had no role in study design, data collection and analysis, decision to publish, or preparation of the manuscript.”

b. If any authors received a salary from any of your funders, please state which authors and which funders.

PGPH-D-23-01242

1. Please provide separate figure files in .tif or .eps format only and remove any figures embedded in your manuscript file. Please also ensure all files are under our size limit of 10MB.

- https://www.sciencedirect.com/science/article/pii/S2211335521001145?via%3Dihub

3. In your revision ensure you cite all your sources (including your own works), and quote or rephrase any duplicated text outside the methods section. Further consideration is dependent on these concerns being addressed.""

4. Please provide additional details regarding participant consent. In the ethics statement in the Methods and online submission information, please ensure that you have specified (1) whether consent was informed and (2) what type you obtained (for instance, written or verbal, and if verbal, how it was documented and witnessed). If your study included minors, state whether you obtained consent from parents or guardians. If the need for consent was waived by the ethics committee, please include this information.

6. In the online submission form, you indicated that "Tha data used to support the findings of this study are available upon request. Reviewers interested in accessing the data can contact us to request access". All PLOS journals now require all data underlying the findings described in their manuscript to be freely available to other researchers, either 1. In a public repository, 2. Within the manuscript itself, or 3. Uploaded as supplementary information.

Additional Editor Comments (if provided):

Reviewers' comments:

Reviewer's Responses to Questions

**Comments to the Author**

1. Does this manuscript meet PLOS Global Public Health’s publication criteria? Is the manuscript technically sound, and do the data support the conclusions? The manuscript must describe methodologically and ethically rigorous research with conclusions that are appropriately drawn based on the data presented.

Reviewer #1: Partly

Reviewer #2: Yes

Reviewer #3: Yes

Reviewer #4: Yes

2. Has the statistical analysis been performed appropriately and rigorously?

Reviewer #1: No

Reviewer #2: Yes

Reviewer #3: No

Reviewer #4: No

3. Have the authors made all data underlying the findings in their manuscript fully available (please refer to the Data Availability Statement at the start of the manuscript PDF file)?

Reviewer #1: No

Reviewer #2: Yes

Reviewer #3: No

Reviewer #4: No

4. Is the manuscript presented in an intelligible fashion and written in standard English?

Reviewer #1: Yes

Reviewer #2: Yes

Reviewer #3: No

Reviewer #4: Yes

5. Review Comments to the Author

Reviewer #1: This manuscript examined the mediating role of resilience in the association between adverse childhood experiences and internet addiction among Tunisian youth. The topic is important and the study has some promise, but there are some major methodological issues that must be considered before any useful conclusions can be made. My specific comments are below

-Introduction: This section does not provide justification for considering resilience as a mediator in the association between ACEs and internet addiction. The cited studies support the role of resilience as a moderator (effect modifier) rather than a mediator. By definition, a mediating variable must be on the causal pathway between the exposure and the outcome. Are there studies that support a causal relationship between ACEs and resilience?

-Methods: This section is significantly lacking in details throughout. More information is needed regarding the study sample (how many were recruited, how many consented, response rate), measurement (how were the main variables used in the analysis?), and statistical methods (see below)

-Statistical methods: No details were provided on the methods used to examine the associations. The authors mention adding the mediator to a model without providing any further details regarding the modeling procedures used.

-How were the numbers in Table 3 obtained?

-In the methods section, there was no mention of adjusting for confounding variables. This is a serious flaw in this manuscript that was not even mentioned in the limitations section.

-Results: In table 4, the coefficients include c' which was not mentioned elsewhere. What is this pathway and how was it obtained?

-In the interpretation of results, the authors make no mention of the direction of the associations and the implications of these directions on the mediation results.

-Discussion: Most of this section is focused on the descriptive results and comparing them to the literature, rather than results that answer the research question (mediation).

-In the discussion, the author's interpretation of the results and their implications is more appropriate for a moderator/effect modifier rather than a mediator. The literature cited in the discussion also supports the role of resilience as a moderator, which would have been a better approach for this analysis than mediation.

-The main limitation, acknowledged by the authors, is the cross-sectional design which does not permit the determination of temporality. This limitation violates the basic assumptions required for mediation analysis, which is another reason why mediation was inappropriate in this case.

Reviewer #2: I enjoyed reading about your research and particularly liked the comparative inclusion of data from multiple countries. I would have liked to see the role of ACE and Resilience on overall behavioral addiction mentioned. I assume you find resilience more significant for males in general addiction or other behavioral addictions. Great work.

Reviewer #3: Information in the abstract does not match with the title, especially the gender difference. Please modify the introduction and method sections of the abstract.

Data should be available online as per plos policy.

Please improve the justification of the study.

Add why this study is needed in the context of the country as similar studies are already published.

Follow the reporting guidelines STOBE and cover all the sections in the STOBE.

Give description of the population.

Add a section for the questionnaire development.

Add a section for the data collection techniques.

Add a section for the bias.

Add a sections for the dependent variables and independent variables.

Add the exclusion and inclusion criteria.

Report the sampling techniques.

Report the sample size calculation.

Repot the validity and reliability of the tools used in this study.

Reviewer #4: “Qualitative variables were

134 represented by absolute and relative frequencies and quantitative ones were represented by

135 means and standard deviations (SD)” why qualitative variables? Is it a mix method study? If so, how can it be express in frequencies?

“To compare percentages and means, we used the Chi

136 square and the Student tests, respectively” mention why and when this test were done and the purpose of the tests.

The gender is covered male and female that is basically sex. Why not author say it sex difference?

The justification of the sex difference should be mentioned in the justification of the introduction.

Report the limitation of the study.

Mention what is new in this study in a section.

Give a recommendation.

What are the further research in needed, add a section for that.

6. PLOS authors have the option to publish the peer review history of their article (what does this mean?). If published, this will include your full peer review and any attached files.

**Do you want your identity to be public for this peer review?** For information about this choice, including consent withdrawal, please see our Privacy Policy.

Reviewer #1: No

Reviewer #2: No

Reviewer #3: No

Reviewer #4: No

---

## [Decision Letter · Decision Letter 1]

7 Nov 2023

PGPH-D-23-01242R1

Gender differences in adverse childhood experiences, resilience and internet addiction among Tunisian students: Exploring the mediation effect.

Dear Dr. Hariz,

Thank you for submitting your manuscript to PLOS Global Public Health. After careful consideration, we feel that it has merit but does not fully meet PLOS Global Public Health’s publication criteria as it currently stands. Therefore, we invite you to submit a revised version of the manuscript that addresses the points raised during the review process.

We look forward to receiving your revised manuscript.

Kind regards,

Humayun Kabir

Academic Editor

Journal Requirements:

1. We noticed you have some minor occurrence of overlapping text with the following previous publication(s), which needs to be addressed:

- https://www.sciencedirect.com/science/article/pii/S2211335521001145?via%3Dihub

In your revision ensure you cite all your sources (including your own works), and quote or rephrase any duplicated text outside the methods section. Further consideration is dependent on these concerns being addressed.""

Additional Editor Comments (if provided):

Reviewers' comments:

Reviewer's Responses to Questions

**Comments to the Author**

1. If the authors have adequately addressed your comments raised in a previous round of review and you feel that this manuscript is now acceptable for publication, you may indicate that here to bypass the “Comments to the Author” section, enter your conflict of interest statement in the “Confidential to Editor” section, and submit your "Accept" recommendation.

Reviewer #1: (No Response)

Reviewer #2: All comments have been addressed

2. Does this manuscript meet PLOS Global Public Health’s publication criteria? Is the manuscript technically sound, and do the data support the conclusions? The manuscript must describe methodologically and ethically rigorous research with conclusions that are appropriately drawn based on the data presented.

Reviewer #1: (No Response)

Reviewer #2: Yes

3. Has the statistical analysis been performed appropriately and rigorously?

Reviewer #1: (No Response)

Reviewer #2: Yes

4. Have the authors made all data underlying the findings in their manuscript fully available (please refer to the Data Availability Statement at the start of the manuscript PDF file)?

Reviewer #1: (No Response)

Reviewer #2: Yes

5. Is the manuscript presented in an intelligible fashion and written in standard English?

Reviewer #1: (No Response)

Reviewer #2: Yes

6. Review Comments to the Author

Reviewer #1: Thank you for addressing my comments. The manuscript has improved significantly.

The only issue that was not corrected is the lack of adjustment for confounders. I was hoping the authors would use regression analysis to control for confounders such as socioeconomic status (parental income or education), family structure (2 parents vs. single parent, number of siblings), anxiety or depression, in addition to age and sex.

Are these variables available in the survey? If one or more variable is available, I recommend using regression analysis to control for measured confounders.

Reviewer #2: Thank you for your response. I hope to see more research done later about this matter

7. PLOS authors have the option to publish the peer review history of their article (what does this mean?). If published, this will include your full peer review and any attached files.

**Do you want your identity to be public for this peer review?** For information about this choice, including consent withdrawal, please see our Privacy Policy.

Reviewer #1: No

Reviewer #2: No

---

## [Decision Letter · Decision Letter 2]

18 Dec 2023

Gender differences in adverse childhood experiences, resilience and internet addiction among Tunisian students: Exploring the mediation effect.

PGPH-D-23-01242R2

Dear Hariz,

We are pleased to inform you that your manuscript 'Gender differences in adverse childhood experiences, resilience and internet addiction among Tunisian students: Exploring the mediation effect.' has been provisionally accepted for publication in PLOS Global Public Health.

Best regards,

Humayun Kabir

Academic Editor

Reviewer Comments (if any, and for reference):

Reviewer's Responses to Questions

**Comments to the Author**

1. If the authors have adequately addressed your comments raised in a previous round of review and you feel that this manuscript is now acceptable for publication, you may indicate that here to bypass the “Comments to the Author” section, enter your conflict of interest statement in the “Confidential to Editor” section, and submit your "Accept" recommendation.

Reviewer #1: All comments have been addressed

Reviewer #2: All comments have been addressed

2. Does this manuscript meet PLOS Global Public Health’s publication criteria? Is the manuscript technically sound, and do the data support the conclusions? The manuscript must describe methodologically and ethically rigorous research with conclusions that are appropriately drawn based on the data presented.

Reviewer #1: Yes

Reviewer #2: Yes

3. Has the statistical analysis been performed appropriately and rigorously?

Reviewer #1: Yes

Reviewer #2: Yes

4. Have the authors made all data underlying the findings in their manuscript fully available (please refer to the Data Availability Statement at the start of the manuscript PDF file)?

Reviewer #1: Yes

Reviewer #2: Yes

5. Is the manuscript presented in an intelligible fashion and written in standard English?

Reviewer #1: Yes

Reviewer #2: Yes

6. Review Comments to the Author

Reviewer #1: Thank you for addressing my concerns. I have no further comments

Reviewer #2: N/A

7. PLOS authors have the option to publish the peer review history of their article (what does this mean?). If published, this will include your full peer review and any attached files.

**Do you want your identity to be public for this peer review?** For information about this choice, including consent withdrawal, please see our Privacy Policy.

Reviewer #1: No

Reviewer #2: No
